# In Silico Designed Multi-Epitope Vaccine Based on the Conserved Fragments in Viral Proteins for Broad-Spectrum Protection Against Porcine Reproductive and Respiratory Syndrome Virus

**DOI:** 10.3390/vetsci12060577

**Published:** 2025-06-12

**Authors:** Shaukat Ullah, Hikmat Ullah, Kainat Fatima, Tan Lei

**Affiliations:** 1Key Laboratory of Quantitative Synthetic Biology, Shenzhen Institute of Synthetic Biology, Shenzhen Institutes of Advanced Technology, Chinese Academy of Sciences, Shenzhen 518055, China; ullah@siat.ac.cn (S.U.); kainat@siat.ac.cn (K.F.); 2University of Chinese Academy of Sciences, Beijing 100049, China; hikmat@siat.ac.cn; 3Center for Energy Metabolism and Reproduction, Institute of Biomedicine and Biotechnology, Shenzhen Institutes of Advanced Technology, Chinese Academy of Sciences, Shenzhen 518000, China; 4State Key Laboratory of Reproductive Medicine and Offspring Health, Nanjing Medical University, Nanjing 211166, China; 5Innovation Center of Suzhou Nanjing Medical University, Suzhou 215000, China; 6National Center of Technology Innovation for Biopharmaceuticals, Suzhou 215000, China

**Keywords:** porcine reproductive and respiratory syndrome virus, PRRSV vaccine, in silico, multi-epitopes, immune response

## Abstract

Porcine reproductive and respiratory syndrome virus (PRRSV) represents a significant threat to the global pig farming industry, primarily due to its high mutation rate and genetic variability, which have hindered the development of effective vaccines. Consequently, we present a groundbreaking approach to vaccine design, using conserved protein fragments from multiple PRRSV strains, afterward predicted optimized epitopes combined that could stimulate B cells, helper T lymphocytes (HTLs), and cytotoxic T lymphocytes (CTLs). Subsequently, our designed vaccine candidate, PRRSV-V-2, demonstrated higher physicochemical properties and effectively bound to immune receptors (TLR3 and TLR8), enhancing immune system activation. Moreover, the recombinant PRRSV-V-2 plasmid, designed for expression in *E. coli*, offers a promising, scalable solution for controlling PRRSV across diverse strains. This novel vaccine could revolutionize swine health management by providing broad-spectrum protection against PRRSV.

## 1. Introduction

Porcine reproductive and respiratory syndrome (PRRS), caused by the PRRS virus (PRRSV), was first identified in the United States in 1987. Since then, PRRSV has become a global threat, inflicting severe economic losses in the swine industry [1]. Recent estimates show that PRRSV losses the swine industry approximately USD 1.2 billion annually [2,3]. In Germany, financial losses due to PRRSV resulted in a 19.1% average reduction in farm profit, reaching up to 41% under severe conditions in 2021 [4]. One of the key challenges in controlling PRRSV lies in its genetic and antigenic variability [5,6,7], which complicates disease tracking and vaccine development. The virus’s high mutation rate has hindered the making of a broadly protective vaccine, prolonging the fight against this devastating pathogen [8,9].

PRRSV is a member of the Arterivirus genus within the Arteriviridae family, with a genome of approximately 15.4 kb and encodes 10 open reading frames (ORFs) [10]. This virus produces two large polyproteins from ORFs 1a and 1b, which are cleaved into 14 nonstructural proteins [11]. Additionally, 7 structural proteins are encoded by ORFs 2–7, including glycoproteins (GP) (GP2a, GP3, GP4, GP5, GP2b or E protein, GP6 or M protein, and GP7 or N protein) [12]. These structural proteins play key roles in virus–host interactions and are highly immunogenic, making them key targets for vaccine development. Currently, 365 distinct PRRSV strains have been isolated worldwide, with significant genetic diversity due to ongoing recombination and mutations in the viral genome. PRRSV is classified into two major genotypes based on these variations: PRRSV1 (European) and PRRSV2 (North American). Notably, the GP2, GP3, GP4, and GP5 proteins are hotspots for mutations, while the GP6 (M protein) and GP7 (N protein) sequences are relatively conserved [13,14]. The emergence of new strains continues to challenge the development of vaccines capable of providing broad protection across diverse genotypes.

In response to this constant evolution, a variety of vaccine types have been developed, including 17 modified live virus (MLV) vaccines, 3 killed virus (KV)vaccines, 5 adenovirus-vectored vaccines, 2 synthetic peptide vaccines, and 14 subunit vaccines, as of 2024 [15,16]. Although MLV and KV vaccines have shown practical effectiveness, each has significant limitations. MLV vaccines are effective against homologous strains but pose safety concerns, such as the risk of virulent reversion and recombination with field strains [17,18,19]. Currently, a total of five MLV vaccines are used in the U.S., seven in Europe, and four in Asia (Appendix A). However, significant safety concerns are always accompanied due to the possibility of virulence reversion and recombination with virulent field strains [19,20]. On the contrary, KV vaccines are valued for safety, and no adverse effects have been reported [21], which benefits from the killed nature and consequently eliminates risks of replication, mutation, or spreading among vaccinated animals. However, it’s regrettable that KV vaccines are often considered less effective than MLV vaccines [17,19], which limits their market capacity (Appendix A). As reported, the enhancement of humoral immune response was constantly disabled with the variety of virus concentrations and inactivation methods [22]. Since then, KV vaccines have not yet been approved in the United States, despite being approved in Europe and other regions.

To provide effective and wide-spectrum immunoprotection, identifying and targeting protective epitopes is crucial in vaccine development. Several neutralizing epitopes have been identified in PRRSV structural proteins, particularly GP3, GP4, GP5, and M [23,24]. GP5, as a major envelope glycoprotein, plays a key role in viral entry and assembly [25], with neutralizing epitopes found in its N-terminus ectodomain [25,26]. Similarly, epitopes in the M proteins have been identified as important for susceptibility to neutralization antibodies [24,27,28]. In M protein, the pattern of GP5, a fragment of 70 amino acid residues, was determined as the crucial modulator of the susceptibility to neutralization antibody in porcine serum [29]. However, partial protection has been observed with immunization using the GP5-M ectodomain, suggesting that additional epitopes are needed for full viral neutralization. Alternatively, neutralizing epitopes have also been shown in GP2, GP3, and GP4 [30,31,32], with another complex formed as an essential factor in receptor binding and viral infection [33,34]. With the progression of epitope discovery, new candidates were continuously identified. For instance, 37–45 amino acids in GP5 of both genotypes [27] and 57–68 amino acids in GP4 of type I [32,35] have been reported as neutralizing epitopes. T cell epitopes were also highlighted from GP4 and GP5 [36,37]. In the N protein, four to five antigenic regions are included [12] and are essential to induce protective B- and T-cell responses [38]. Despite these findings, cross-genotype protection remains a major hurdle due to the high mutation rate of PRRSV antigens, which allows for immune evasion and complicates vaccine development. Optimized epitopes must be identified to update the vaccine formulation and address this challenge.

To discover effective epitopes with high efficiency in current vaccine strategies, an in silico approach is proposed for identifying novel epitopes and designing a multi-epitope vaccine. With the current advancement of immunoinformatics, the identification of epitopes and formulation of vaccines has become feasible and efficacious [39]. Under the in silico strategy, vaccines have been designed successfully for pathogens, such as the human coronavirus [40], dengue virus [41], influenza virus [42], etc. In the current study, we conducted an in silico design to target either type of PRRSV. The effective and non-risky epitopes were screened from the viral protein fragments conserved across the globally isolated strains. Subsequently, the vaccine candidate was constructed by assembling selected epitopes with different adjuvant proteins. With the evaluation of candidates in physiochemistry, structural biology, and immunology, one promising candidate was ultimately highlighted. This study aims to provide a promising strategy for overcoming the challenges of PRRSV vaccine development, offering cross-genotype protection despite the genetic diversity of PRRSV.

## 2. Materials and Methods

As illustrated in Figure 1, the PRRSV vaccine design was carried out stepwise in three stages. In the first stage, viral protein sequences were retrieved worldwide and then used to extract the conserved fragments inside as the basis of epitope screening. The second stage aimed to obtain the candidate vaccine, including the steps of epitope primary screening, evaluation, selection, and assembly. For the third and last stage, candidate vaccines were subjected to a panel of evaluations to visualize their properties in physiochemistry, structure, host interaction, and immune induction capacity. The web links of databases and online platforms used in this study were listed in Appendix A. All the Supplementary Tables are included in Appendix A, and all the figures are included in Appendix A.

### 2.1. Retrieval of PRRSV Protein Sequence

To initiate the design, full-length sequences of all nonstructural and structural proteins from both PRRSV genotypes (PRRSV1 and PRRSV2) were retrieved from NCBI (Appendix A), focusing on sequences deposited between 2018 and 2023. Conserved fragments within each protein were extracted using multiple sequence alignment (MSA). Initial alignment was conducted using MUSCLE [43], followed by confirmation with CLUSTALW version 2.0 [44]. Conserved sequences shared across retrieved strains were extracted using BioEdit version 7.2 [45] and are listed in Appendix A.

### 2.2. Step-by-Step Selection of Epitopes

Identification of epitopes for B cells, HTLs (helper T cells), and CTLs (cytotoxic T cells) was carried out using IEDB (Immune Epitope Database) tools based on the conserved protein fragments in each viral protein.

The Bepipred Linear Epitope Prediction 2.0 method of IEDB was applied for linear B-cell epitopes. The minimum length of the epitope was set as 6 amino acids. The threshold of 0.5 was used to determine the likelihood of an immunogenic epitope [46]. The NetMHCIIPan 4.1 EL recommended method of IEDB was employed to predict HTL epitopes. Since there were no SLA options for MHCII prediction, the HLA reference was set as below: HLA-DPA1*02:01/DPB1*05:01, HLA-DRB4*01:04, HLA-DRB1*12:04, HLA-DPA1*02:01/DPB1*04:01, HLA-DRB5*01:03, HLA-DPA1*01:03/DPB1*01:01, HLA-DPA1*02:01/DPB1*02:01, HLA-DRB1*04:04) [47]. A fixed length of 15 amino acids was used in the prediction of HTL, and the top 10% of affinity was considered for further analysis, among which %Rank < 2% were regarded as strong binders (SB) and %Rank < 10% for weak binders (WBs). For the prediction of CTL epitopes, the NetMHCpan EL4.1 recommended method of IEDB was used with an MHCI source specified as “Pigs”. Epitope lengths were defined as 9 amino acids, targeting the most common SLA (swine leukocyte antigen) class I alleles [48]: SLA-1*0702, SLA-1*1101, SLA-2*0101, SLA-2*1101, SLA-2*1201, SLA-3*0101, SLA-3*0401, SLA-3*0501, and SLA-3*0701. For CTL peptides ranked within the top 1% (%Rank < 0.5% were considered as strong binders (SB), and %Rank < 2% thresholds are considered for weak binders (WBs)), affinity was selected for further analysis [47].

Selected epitopes for all three types of B cells, HTLs, and CTLs were further assessed for conservancy, antigenicity, allergenicity, and toxicity with the IEDB analysis resource [49], VaxiJen v2.0 bioinformatics tool for protein antigenicity prediction with a threshold of 0.4 for viruses that excludes non-antigen epitopes from vaccine design [50], AllerTOP v.2.0 bioinformatics tool for protein allergy prediction that excludes allergenic epitopes from vaccine design [51] and the ToxinPred bioinformatics tool for protein toxicity prediction that excludes toxic epitopes from vaccine design [52], respectively. The inclusion criteria required sequence identity ≥80%, antigenicity score >0.5, and non-allergic, non-toxic properties. Additional scoring of immunogenicity was conducted for CTL epitopes with the Class I Immunogenicity server of IEDB [49], and the ones who scored ≤0.2 were excluded. For HTL epitopes, further predictions were made to identify potential inducers of IFN-γ and IL-4 using the IFNepitope [53] and IL4pred [54] servers.

### 2.3. Design and Assembly of Candidate Vaccines

The candidate epitopes that passed the stepwise selection process were then used for vaccine construction. Initially, the epitopes were categorized based on their reactivity with B cells, HTLs, and CTLs, and subsequently assembled into a core antigen. Distinct adjuvant domains were attached to enhance the immune response, as shown below:

S50 L7/12 ribosomal protein, isolated from *Mycobacterium tuberculosis*, is known for its ability to provoke an immune response [55].

β-defensin, recognized for its antimicrobial and immunomodulatory properties [56].

Heparin-binding hemagglutinin (HBHA), sourced from *Mycobacterium* species, is known for its immunogenic properties [57].

For vaccine assembly and further reinforcement of immunogenicity, the following linkers were used to organize the immune elements [55]:KK: linker to connect B-cell epitopes [58].AAY: linker used for the organization of CTL epitopes [59].GPGPG: linker to group HTL epitope [41].EAAAK: a rigid linker to fuse the adjuvant domains to the N-terminus of the core antigen.

### 2.4. Evaluation of Candidate Vaccines in Physicochemical and Immune Properties

The Expasy ProtParam Tool [60] was employed to predict a range of physicochemical properties for the candidate vaccines, including molecular weight, isoelectric point, half-life, stability, and other essential characteristics. A protein whose instability index is smaller than 40 is predicted as stable. The solubility of these molecules was predicted using SOLpro of SCRATCH [61]. A threshold of >0.5 confirms that the chosen candidates remained soluble and functional under physiological conditions. Additionally, the immune properties of the candidate’s vaccine were assessed using immunoinformatic tools, including VaxiJen v2.0 for antigenicity, AllerTop for allergenicity assessment, and ToxinPred to evaluate toxicity.

### 2.5. Prediction and Validation of Molecular Structure for Candidate Vaccines

The secondary structure of the vaccine molecules, including α-helices, β-strands, and coils, was predicted using the PSIPRED server [62] based on their polypeptide sequences.

Three-dimensional (3D) structures of the designed vaccine construct were predicted using the PHYRE2 server [63], a widely used platform for protein modeling and analysis. With PHYRE2, you can construct 3D models, predict ligand-binding sites, and evaluate the effect of amino acid variants on protein function and structure. With a confidence level of 90%, the model demonstrated high-confidence structural predictions suitable for downstream structural and immunological analyses. The predicted model by PHYRE2 was further refined with the Galaxy Refine server [64]. Five revised models were suggested for each candidate based on GDT-HA, RMSD, MolProbity, Clash Score, Poor Rotamers, and Rama Favoritism. Subsequently, to validate the refined models, ProSA-web [65] and PROCHECK [66] were employed, generating a Z-score and Ramachandran plot, which were used to assess the overall quality of vaccine proteins.

### 2.6. Molecular Docking and Dynamic Analysis of Vaccine-TLR (Toll-like Receptors) Complexes

To evaluate the potential recognition of vaccine molecules by the host immune system, molecular docking was performed between the candidate vaccines and two pattern recognition receptors (TLRs), TLR3 (PDB ID: D0V2A2) and TLR8 (PDB ID: Q865R7) of swine were retrieved from the UniProt protein data bank [67,68]. HDOCK [69] was used for docking, which uses a hybrid algorithm to combine template-based modeling and free docking. In the docking procedure, HDOCK used its default rigid-body protocol and ranked and refined docked conformations through iterative global as well as local search optimization. The multi-receptor docking approach provided a comprehensive assessment of the vaccine constructs’ potential to engage key pattern recognition receptors and trigger innate immune responses. For each vaccine–TLR complex, model 1 with the lowest binding energy was selected for analysis of non-covalent interactions, including hydrogen bonds, salt bridges, and interface areas, using the PDBsum [70] server. The 3D structures with related interactions were then visualized by PyMOL [71].

For the top-ranked docking models, PRRSV vaccine with -TLR complexes, molecular dynamics (MD) simulations were performed using the AMBER [72]. A 14-angstrom padding around the protein and water box was applied. The AMBERS FF14SB force field [73,74] was used for both vaccines and TLRs. The vaccine molecules were placed in a TIP3P water box with counterions added to neutralize the system. To minimize structural collisions, steepest descent or conjugate gradient minimization methods were employed, with transition steps every 1000 cycles up to 5000 cycles. Thermal equilibrium was reached at 300 K after 250 picoseconds (ps) of system heating. A production run of 100 nanoseconds (ns) was conducted to explore the dynamic behavior of the vaccine–receptor complexes. In the canonical ensemble, periodic boundary conditions were applied to maintain the rigidity of the covalently bound hydrogen atoms using the SHAKE algorithm [75]. In these simulations, the temperature was regulated at 300 K using a Langevin thermostat with a 10 Å cutoff for non-bounded interactions and Ewald simulations for long-range interactions [76]. Statistical parameters such as the root mean square deviation (RMSD), root mean square fluctuations (RMSF), and radius of gyration (Rg) of vaccine–receptor complexes were analyzed using the CPPTRAJ module [77]. The results were visualized in PyMOL [78] to assess the structural stability and dynamic properties of the vaccine-receptor.

### 2.7. Immune Simulation of Candidate Vaccines

Immune simulations were conducted using the C-ImmSim server [79] to model the immune responses elicited after vaccine administration according to a specified immunization schedule. The simulation protocol involved three doses of immunization with a two-week interval, corresponding to simulation steps 1, 84, and 168, where each step represented eight hours of real time. Default settings were used for all additional parameters to ensure consistency and standardization. The dynamics of innate and immune response properties were visualized as curves.

### 2.8. In Silico Cloning of Selected Vaccine Candidate

To generate the DNA coding sequence of the vaccine protein, JCat servers were used for reverse translation and codon optimization based on *E. coli* codon usage bias. The codon adaptation index (CAI) and GC content percentage of the resulting sequence were calculated to assess the protein expression level [80]. A good outcome was defined as a CAI > 0.8, while GC content ranged from 30% to 70%. Furthermore, in silico cloning was performed in SnapGene 8.0.2 using the pET-28a (+) vector with restriction sites Xhol and BamHI.

## 3. Results

### 3.1. Extraction of Conserved Fragments from PRRSV Protein Sequences Collected Worldwide

Due to the nature of rapid evolution, PRRSV protein sequences deposited from 2018~2023 were taken as the starting material for the current design. Approximately 7315 full-length sequences were downloaded in total from NCBI in FASTA format, including 90 proteins for PRRSV1 and 7225 for PRRSV2. Protein accession numbers are listed in Appendix A. Based on the alignments, conserved fragments were identified for each strain, and those with at least 10 amino acids were used for epitope screening. Finally, 30 conserved fragments were obtained for PRRSV1 and 77 for PRRSV2 (Appendix A).

### 3.2. Prediction and Evaluation of Epitopes for B Cells, CTLs, and HTLs

B-cell epitope prediction was performed based on structural proteins of PRRSV, including GP2a, GP3, GP4, GP5, M, and N. Primarily, 68 epitopes were generated for PRRSV1 (European genotype) and 97 for PRRSV2 (American genotype). Epitopes were then evaluated for their antigenicity, allergenicity, and toxicity. Subsequently, the ones that were antigenic, non-allergic, and non-toxic were reserved for further usage, including 06 for PRRSV1 and 07 for PRRSV2. The conservancy of epitopes was also analyzed using the epitope conservancy analysis algorithm in the IEDB platform, and the ones scored above 50% were highlighted among the reserved epitopes. The final selection of B-cell epitopes that were present on the surface of each protein is included in Table 1. The procedure of epitope selection is presented in Appendix A and Appendix A.

Epitopes for T lymphocytes were predicted from all PRRSV proteins, and 4560 epitopes were generated for HTLs and 4208 for CTLs (Appendix A). The first selection was then carried out based on binding affinity between candidate epitopes and MHC II or I alleles, with the top 10% and the top 1% as the threshold, respectively [81]. Further selection of epitopes was conducted for both HTLs and CTLs to highlight the most promising inducer of T cell reaction. For CTLs, epitopes were chosen once antigenicity scored >0.5 and immunogenicity > 0.1, non-allergenic and non-toxic, while excluded if allergenic, toxic, or conservancy was lower than 50% (Appendix A). Similar filters were also set up for HTLs, but the capacity of IFN-γ and IL-4 induction was evaluated instead of immunogenicity scoring (Appendix A). Resultantly, 10 HTL epitopes and 10 CTL epitopes were selected for the final vaccine design (Table 2 and Table 3).

### 3.3. Design of Three Candidate Vaccines

To construct the candidate vaccines, the core antigen was first organized via the assembly of selected epitopes for B cells, HTLs, and CTLs, with the help of linker sequences. KK, GPGPG, and AAY linkers were employed to group three classes of epitopes (Figure 2A). After that, three distinct adjuvant proteins, L7/12 S50 ribosomal protein, β-defensin, and HBHA, were added to enhance the boosting of the host immune system. The adjuvant proteins were attached with the core antigen at the N-terminus with EAAAK, a rigid linker. Correspondingly, three candidate vaccines were generated and termed PRRSV-V-1, PRRSV-V-2, and PRRSV-V-3, as shown in Figure 2B–D, while protein sequences of vaccine candidates were contained in Appendix A:PRRSV-V-1: Core antigen attached with S50 L7/12 ribosomal protein on N-terminal.PRRSV-V-2: Core antigen attached with β-defensin on the N-terminal.PRRSV-V-3: Core antigen attached with HBHA adjuvant on N-terminal.

### 3.4. Properties of Candidate Vaccines in Physicochemical and Immunology

To ensure the qualification of candidate vaccines primarily, their antigenicity, allergenicity, and toxicity were analyzed before any other characteristics (Table 4). The physicochemical properties were analyzed thereafter to assess their molecular size, stability, solubility, and hydropathicity, which were correlated with the feasibility of the future preparation and application of candidate vaccines. For the molecular size, all three candidate vaccines, PRRSV-V-1~3, contained approximately 531 amino acids, 448 amino acids, and 551 amino acids (line 1 in Table 4) and weighed around 57 kDa, 49 kDa, and 60 kDa (line 2 in Table 4). To ensure good stability in vitro, instability indexes were calculated much lower than 40 (line 3 in Table 4). The vaccines’ aliphatic index ranged between 69 and 82, indicative of a stable protein (line 4, Table 4). For intracellular stability, the protein half-life was estimated to be more than 30 h in mammalian cells, beyond 20 h in yeasts, and in *E. coli*, no less than 10 h (line 5 in Table 4). The water solubility of candidates was scored with SOLpro. Resultantly, the solubility scores were around 0.9 (line 6 in Table 4), GRAVY slightly below 0 (line 7 in Table 4), and pI generally above 9 (line 8 in Table 4). In one word, these three candidate vaccines were suggested as hydrophilic and soluble in water at physiological pH. All three molecules showed non-allergenicity and non-toxicity and scored >0.6 in antigenicity (lines 9–11 in Table 4), which is deemed to be safe and effective in immunization.

### 3.5. Prediction and Refinement of Secondary and Tertiary Structure

To determine the composition of secondary structural elements in candidate vaccines, the PSIPRED server was used (Figure 3A–C). PRRSV-V-1 contains 44.06% of alpha helix, 11.11% of beta-strand, and 44.82% of random coil, as plotted along its polypeptide sequence. The alpha helix, beta-strand, and random coil in PRRSV-V-2 were 43.30%, 12.05%, and 44.64%, while 57.16%, 6.71%, and 36.11% in PRRSV-V-3, respectively (Figure 3D).

For the modeling of tertiary structure, templates c8yeqA, c1kj6A, and c2ch7B were employed on the Phyre2 platform corresponding to PRRSV-V-1, 2, and 3 candidates (Figure 4A–C). The levels of confidence and coverage are shown in Figure 4D,E.

The structural models were then verified with Procheck, and the Ramachandran plots were generated subsequently. As visualized in the Ramachandran plot, 96.7% of residues in PRRSV-V-1 were located in the favored region, 3.3% in the allowed region, and 0% in the disallowed region. Similarly, the residues of PRRSV-V-2 and 3 were mainly in the favored region, which is 84.2% and 99.4%, and the allowed region was 15.8% and 0.6%, respectively (Figure 5A–D). On the other side, ProSA-web showed Z-scores of −5.98, −4.80, and −1.24 for vaccine candidates PRRSV-V-1, 2, and 3, respectively, with which high quality was indicated for all three models (Figure 5E–H).

### 3.6. Molecular Docking and Dynamics Simulation for Complexes of Vaccine-Immune Receptor

Molecular docking was performed using the HDock server to evaluate the interaction between the vaccine protein and pattern recognition receptors. Two receptor molecules of swine, TLR3 and TLR8, were used in docking with three candidate vaccines, and then 6 complexes were generated. For each complex, 10 models were constructed and ranked by docking score from low to high, among which model 1 exhibited the strongest binding.

Tight binding between vaccines and receptors was suggested, with the lowest binding energy score ranging from −100~−350 kJ/mol (Appendix A). The interactions were illustrated in detail, while the number of hydrogen bonds, salt bridges, non-ionic interactions, and the size of the interface were listed. To PRRSV-V-1, complexes with TLR3 and 8 were scored as −196.74 and −190.43 kJ/mol, respectively. The interface between PRRSV-V-1 and TLR3 was composed of 15 and 20 residues on either the vaccine or receptor side, respectively. The area of this interface reached 1104 Å2, within which 3 hydrogen bonds, 2 salt bridges, and 121 non-covalent interactions were formed. An interface of a much smaller scale was formed between PRRSV-V-1 and TLR8, which contained a much smaller number of connections, including 2 hydrogen bonds, 0 salt bridges, and 87 non-covalent interactions. PRRSV-V-2 exhibited strong and consistent binding across TLR3 and TLR8. It achieved the best docking score with TLR3 (–308.16 kJ/mol) and TLR8 (−263.17 kJ/mol), forming a stable interface with 6, 9 hydrogen bonds, 3, 1 salt bridges, and 155,176 non-covalent interactions, as illustrated in Figure 6. The other PRRSV-V-3 docking analysis showed strong interactions (−260.99 kJ/mol with TLR3 and −335.61 kJ/mol with TLR8), along with a stable interface with 3,3 hydrogen bonds, 4, 1 salt bridges, and 316,282 non-covalent interactions. All these docking analyses of PRRSV-V-1~3 are presented in Table 5. PRRSV-V-1 and PRRSV-V-3 docking interactions, number of hydrogen bonds, salt bridges, and non-covalent interactions were supplemented in Appendix A. Molecular docking analysis revealed that PRRSV-V-2 and PRRSV-V-3 bind most favorably to innate immune receptors TLR3 and TLR8, resulting in significantly lower docking scores and enhanced interfacial interactions. Based on these results, their potential as highly immunogenic vaccine candidates is highlighted by the likelihood of receptor engagement and downstream immune activation. In contrast, PRRSV-V-1 exhibited significantly lower binding affinity and fewer interaction interfaces, suggesting a limited ability to trigger an immune response.

The conformational stability and dynamic behavior of vaccine–receptor complexes were further evaluated using molecular dynamics simulations (MD). A complex’s overall structural stability was determined by RMSD, while its compactness was determined by Rg. In the PRRSV-V-2–TLR3 complex, the RMSD value consistently remained within an acceptable range below 4, while in the PRRSV-V-2–TLR8 complex, equilibrium was reached approximately 40 ns after reaching equilibrium, indicating that conformational behavior had remained stable throughout the simulation (Figure 7A). The Rg values for PRRSV-V-2–TLR3 and PRRSV-V-2–TLR8 complexes range from 32.0 to 33.5, indicating compact and stable conformations throughout simulation (Figure 7B). Furthermore, the number of intermolecular hydrogen bonds fluctuated between 8 and 16 in the complexes of PRRSV-V-2 with TLR3 or TLR8; therefore, persistent and favorable vaccine–receptor interactions were suggested (Figure 7C). However, with overranged RMSD and Rg indicated during simulation, PRRSV-V-1 and PRRSV-V-3 molecules were suggested to be unable to form stable complexes with TLR3 and TLR8. RMSD values of the PRRSV-V-1–TLR3 and PRRSV-V-1–TLR8 complexes exceeded 5 Å and 30 Å, respectively. There were also pronounced fluctuations in the radius of gyration (Rg) of PRRSV-V-1 and PRRSV-V-3 with TLR3/TLR8 complexes, ranging from 54 Å to 34 Å, suggesting partial unfolding and loss of structural compactness (Appendix A). Based on the above analysis and comparison with PRRSV-V-1 and PRRSV-V-3, such variability of key structural parameters implies weakened and less stable interactions. Based on the analysis above, the variability of key structural parameters among the three candidate vaccine molecules suggested PRRSV-V-2, the one with robust interactions with TLR3 and TLR8.

### 3.7. Simulation of Immune Responses Post-Vaccination

To demonstrate the efficacy of vaccine candidate PRRSV-V-2 to induce host immunity, a simulation was then conducted on the platform C-ImmSim with three doses of vaccination at 4-week intervals till 350 days post-immunization. Dendritic cells, macrophage cells, and natural killer cells were monitored in the simulation to see the status of innate immunity. A significant activation of macrophage cells was demonstrated following each dose of immunization. In addition, cytokines like IFN-γ and IL-2 were also rapidly elevated up to 400,000 and 700,000 ng/mL, respectively. Thus, it can be seen that vaccine candidate PRRSV-V-2 can initiate the innate part of the host immune system, as shown in Figure 8.

The adaptive immune response was also analyzed in this simulation to assess the activation of B cells, T cells, and immunoglobulins, with a particular focus on the B- and T-cell-related immune memory.

Following three vaccine doses, a robust and long-lasting activation of B cells was observed, with notable increases in memory B cells and IgM isotypic B cells, which persisted till one year after the initial immunization. Production of antibodies was observed in a pattern following the dynamics pattern corresponding to B cell activation. The total levels of IgM and IgG peaked at nearly 200,000, with individual levels reaching around 100,000 for each, suggesting a potent humoral response comparable to other published vaccines designed in silico [41]. T-cell activation was also evident, with strong responses in both cytotoxic and helper T-cell populations. Notably, memory helper T cells displayed high dynamic activity, maintaining a level above 600 at 350 days post-immunization, corresponding to 30% of the peak response. The peak immune response across different parameters was achieved approximately 50 days after the first dose, coinciding with the period immediately following the third vaccination (Figure 9).

These findings highlight the vaccine’s robust immunogenic potential, suggesting that it can induce strong and systematic adaptive immune protection against PRRSV infection. Especially, prolonged memory B- and T-cell responses indicate the potential for long-term immunity following immunization.

### 3.8. Codon Optimization of the Final Vaccine Candidate

Its coding sequence was first generated using the JCat server to produce the designed vaccine. Codon optimization was then performed with JCat to align the codon usage bias of *E. coli* K12. The optimized DNA sequence, consisting of 1386 bp, had a GC content of 56.02% and a 0.90 CAI, indicating efficient expression potential. This sequence was subsequently cloned into the pET-28a (+) vector between *XhoI* and *BamHI* restriction sites. As a result, a recombinant plasmid for vaccine expression was successfully constructed, ready for application in the *E. coli* expression system, as shown in Appendix A.

## 4. Discussion

Porcine reproductive and respiratory syndrome (PRRS) has been a significant veterinary disease for over three decades, causing substantial economic losses and affecting pig farms worldwide. Despite extensive research, specific therapeutic drugs remain ineffective, and current vaccines offer only limited protection. Moreover, current commercially available vaccines offer just modest protection. Even worse, among the technical platforms ever tested, only modified live virus [19] or killed virus [18]-based vaccines have been proven to be effective. These challenges have made the control of PRRS exceptionally difficult.

Several key issues complicate the development of an effective PRRSV vaccine [82]. First, the extensive genetic variations in PRRSV genomes complicated the host immune response, making it difficult to achieve full protection across strains, which is only achievable with strains homologous to the vaccine [83]. Second, PRRSV is prone to genetic recombination and replacement, particularly among RNA viruses. This poses a substantial risk of virulence reversal or genetic recombination with field strains when live attenuated vaccines are used [84]. Although inactivated vaccines can mitigate this risk, they do not effectively promote cellular immunity. Furthermore, certain viral proteins interact with the host immune system, evading both innate and adaptive immunity, which either prolongs infection or reduces the immunogenicity of vaccines [85]. Several neutralizing epitopes have been identified in PRRSV structural proteins, particularly GP3, GP4, GP5, and M [23,24]. GP5, as a major envelope glycoprotein, plays a key role in viral entry and assembly [25], with neutralizing epitopes found in its N-terminus ectodomain [25,26]. Similarly, epitopes in the M proteins have been identified as important for susceptibility to neutralization antibodies [24,27,28]. In M protein, the pattern of GP5, a fragment of 70 amino acid residues, was determined as the crucial modulator of the susceptibility to neutralization antibody in porcine serum [29]. However, partial protection has been observed with immunization using the GP5-M ectodomain, suggesting that additional epitopes are needed for full viral neutralization. Alternatively, neutralizing epitopes have also been shown in GP2, 3, and 4 [30,31,32], with which another complex is formed as an essential player in receptor binding and viral infection [33,34]. Thus, in-depth investigation and continuous development of new vaccines with great potency will provide targeted opportunities for preventing and controlling PRRSV. However, a long way of struggle is still required to bring forth the vaccine design.

With the advancement of immunoinformatic platforms, a new path for vaccine development has emerged, distinct from the traditional approaches. Instead of relying on a small number of representative strains in the wet lab, large panels of strains from the same species can be utilized to identify epitopes. This approach can enhance the quality and coverage of vaccines. By predicting and scoring epitopes using various bioinformatics tools, promising epitopes can be selected, while suboptimal or risky ones are excluded. Subsequently, this method enables the quick construction of candidate vaccines by combining selected epitopes with adjuvant proteins, followed by multidisciplinary evaluations. Based on the predicted structural and immunogenic properties, the range of optimized proposals is narrowed down from a panel of candidates before conducting actual wet lab assays. This approach has facilitated the de novo design of vaccines for a wide range of pathogens, particularly those with significant complexity and challenges in vaccine development. With the benefit brought out with such an effective strategy, the previous design of the PRRSV vaccines in 2024 and 2025 dates [81,86]. The studies acknowledged certain limitations that need further investigation. In our study, a multi-epitope vaccine was developed to protect the swine from either PRRSV1 or 2, eliciting both humoral and cellular immune responses. In this vaccine, epitopes were screened from 30 and 77 conserved fragments of viral proteins in PRRSV1 and PRRSV2, respectively, and 6, 10, and 10 epitopes were selected to target B cells, CTLs, and HTLs. Stringent criteria based on antigenicity, immunogenicity, allergenicity, toxicity, conservancy, and the capacity of cytokine induction were applied. Three candidate vaccines were generated by organizing the selected epitopes and conjugating them with adjuvant proteins such as S50 L7/12 ribosomal protein, β-defensin, and HBHA. In silico evaluations demonstrated good structural properties, tight interactions with host pattern recognition receptors, and strong immune stimulation. PRRSV-V-2 emerged as the most suitable candidate due to its favorable interaction with TLR3 and TLR8. The MD-derived structural stability metrics also have direct implications for vaccine effectiveness. For sustained receptor engagement and effective innate immune activation, vaccine–TLR interactions must be stable. Most residues of the complex exhibit a few atomic fluctuations, particularly at key contact regions, as indicated by the low Rg values across most residues. As a result of this structural rigidity, receptor engagement may be consistent, leading to reliable downstream signaling and cytokine activation, resulting in an enhanced ability to elicit a strong and sustained adaptive immune response. To effectively defend against viruses, vaccine constructs that interact with TLRs may enhance Th1-skewed immune responses [87].

When compared to earlier in silico PRRSV vaccine design [81,86], our approach differed significantly. While the previous design was based on pathogenic isolates specific to his research group, our study included PRRSV protein sequences from global strains, providing a broader representation. Additionally, we used all viral proteins, focusing on conserved fragments across strains, whereas Dr. Liu’s design was restricted to GP3 and GP5 molecules. This difference potentially offers a wider range of epitope sources for our design and potentially improves the vaccine’s coverage in a broader array of PRRSV strains. At the next stage, screening of epitopes was carried out in previous work; however, the evaluations of allergenicity and toxicity were absent in the earlier design [81,86]. In speculation, these were due to the limited supply of source antigens; however, potential risks were then introduced. Unlike that, with plenty of sources of viral antigens, enough resultant epitopes were still selected in our work, even after a stringent evaluation of risks to each single epitope predicted originally. With the abundance of outcomes, more CTL and HTL epitopes were selected to construct the vaccine candidates with a reasonable length sequence of 448 amino acids, as compared to the previous PRRSV in silico design vaccines’ lengths of 688 and 455 amino acids [81,86]. Possibly, this will be a positive factor to reinforce the T-cell immunity, which is equally important in the control of disease and restraint of severity post-infection. In the stages, after epitopes were obtained, more adjuvant proteins and pattern recognition receptors were tried in our work. Especially, rather than human TLR3 and TLR4 used in previous studies [81,86], docking of our candidate vaccines was performed with TLR3 and TLR8 of swine, the prospective host. This approach offers more accuracy in predicting the real-world interaction between vaccines and receptors in the host. Additionally, our immune simulation was extended to 350 days, rather than 35 days in Dr. Liu’s study, which provided a more comprehensive understanding of immune persistence. With longer observation, the persistence of immune protection was demonstrated in candidates of our selection. The scale of total and active B cell populations was maintained at a high level for up to one year, while the production of immunoglobulins was also retained to a certain degree. Notably, about 30% of peak levels were reserved for the memory B and T cells until the end of the simulation, consolidating the confidence in our strategy and choice in the current design.

## 5. Conclusions

By using comprehensive in silico approaches, we designed a novel multi-epitope vaccine candidate to address PRRSV’s high mutation rate and genetic variability. We identified conserved antigenic regions and systematically screened them for B cell, helper T lymphocyte (HTL), and cytotoxic T lymphocyte (CTL) epitopes by integrating genomic and proteomic data from multiple PRRSV strains. Epitopes were selected based on antigenicity, immunogenicity, non-allergenicity, and non-toxicity criteria. Among the three vaccine candidates evaluated, PRRSV-V-2 displayed acceptable physicochemical properties, stable interactions with TLRs, and consequently, robust induction of host immunity. In addition, an expression plasmid for PRRSV-V-2 was constructed in silico, facilitating downstream studies.

Using computational tools for high-throughput epitope identification and validation, our approach streamlines vaccine development at the early stage. Through molecular docking and molecular dynamics simulations, we gained further insight into the conformational stability and binding affinity of vaccine–receptor complexes, increasing our confidence in the vaccine candidate’s ability to elicit robust immune responses. With this strategy, time, cost, and resource burden can be significantly saved when compared with traditional experimental approaches. Even so, this study’s findings are inherently predictive and limited by in silico methodologies. Although computational tools are powerful, they cannot capture the immunological complexity of a living host or replicate the dynamic interaction between immune cells and signaling molecules. It is therefore necessary to experimentally validate the predicted immunogenicity, safety, and efficacy of the proposed vaccine construct.

To evaluate cellular immune responses, future directions will involve expressing and purifying the PRRSV-V-2 construct in vitro in a bacterial expression system, followed by immunological assays, including ELISpot, T-cell activation profiling, and cytokine release tests. In vivo studies will be essential to assess protective efficacy, immunological memory induction, and overall safety. Experimental validation will be crucial for translating our computational findings into a preclinical vaccine candidate with real-world application potential.

## Figures and Tables

**Figure 1 vetsci-12-00577-f001:**
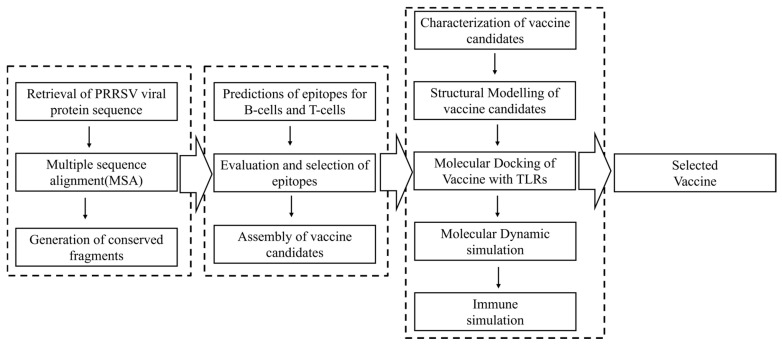
An overview of the reverse vaccinology process used to design the Porcine reproductive and respiratory syndrome virus vaccine. Data collection, epitope prediction, vaccine candidate construction, and in silico validation are highlighted.

**Figure 2 vetsci-12-00577-f002:**
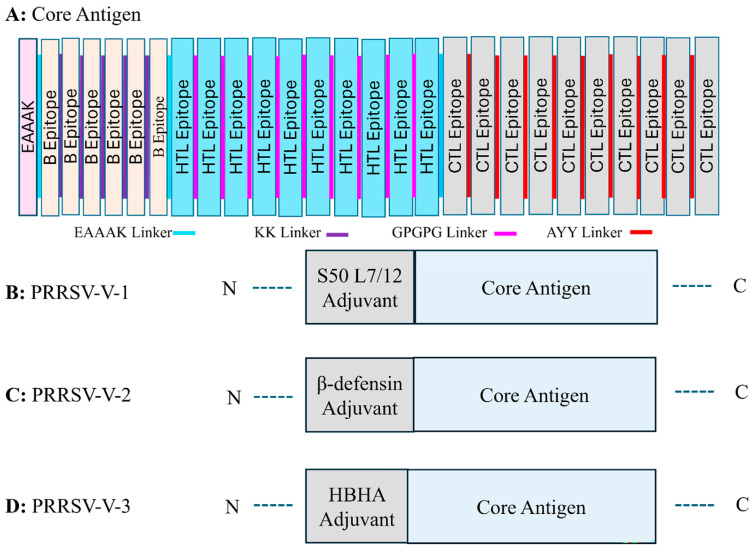
Organization of multi-epitope vaccine peptides. (**A**) Graphic representation of the vaccine design approach that includes the core antigen and a summary of the approach taken to create the multi-epitope vaccine peptide. This design includes specific B cells, CTLs, and HTLs epitopes that are bordered by adjuvant and connected by peptide linkers. (**B**–**D**) Vaccine candidates of PRRSV-V-1, PRRSV-V-2, PRRSV-V-3 (**B**), PRRSV-V-1, Core antigen conjugated with S50 L7/12 ribosomal protein at the N-terminal, (**C**) PRRSV-V-2, at the N-terminal, β-defensin is attached to the core antigen, (**D**) HBHA adjuvant is conjugated with core antigen at the N-terminal.

**Figure 3 vetsci-12-00577-f003:**
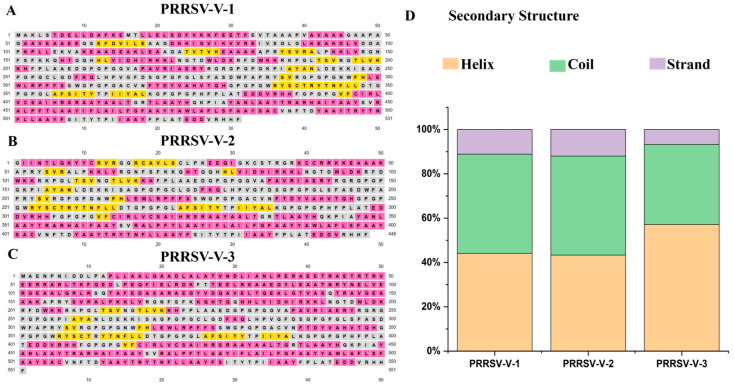
Secondary structure of vaccine candidate. (**A**–**C**) Illustration of the secondary structure for the PRRSV-V-1~PRRSV-V-3 vaccine candidate. Strand, helix, and coil are indicated with distinct colors: yellow, pink, and gray, respectively. (**D**) Histogram plot of secondary structure for PRRSV-V-1~PRRSV-V-3, showing the percentage distribution of coils, helices, and β-strands in each protein.

**Figure 4 vetsci-12-00577-f004:**
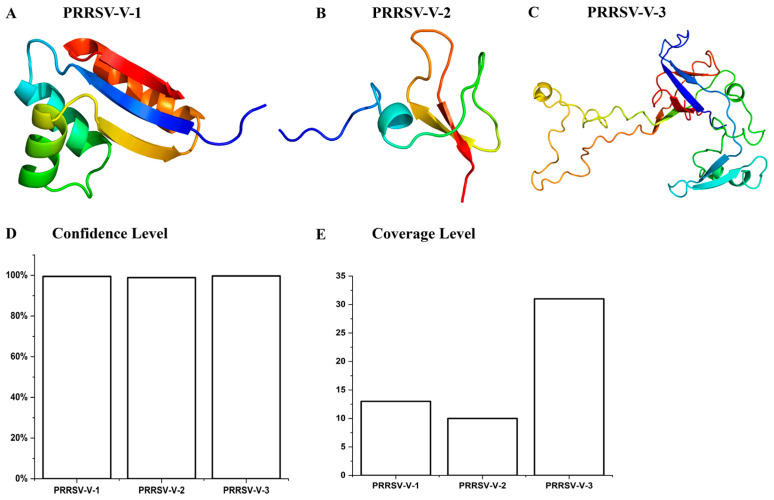
Tertiary structure of vaccine candidate. (**A**–**C**) The tertiary structure of the template model was used to model the PPRRSV-V-1~PRRSV-V-3 tertiary structure. (**D**) Confidence level for the tertiary structure models of PRRSV-V-1~PRRSV-V-3. (**E**) Coverage level of tertiary structure models PRRSV-V-1~PRRSV-V-3.

**Figure 5 vetsci-12-00577-f005:**
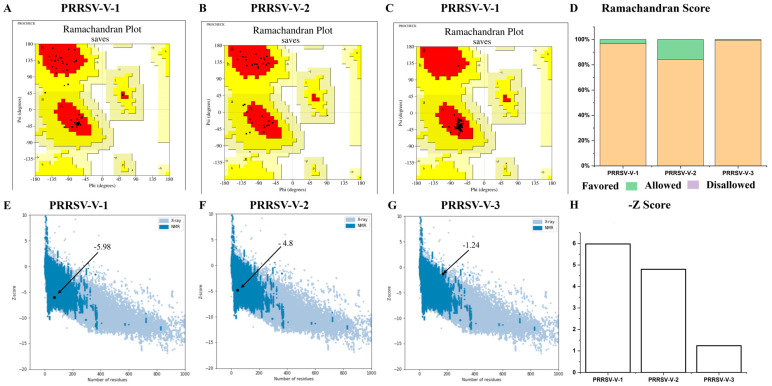
Validation of vaccine candidates’ molecular structure. (**A**–**C**) The Ramachandran plot of PRRSV-V-1~PRRSV-V-3 emphasizes the residues in favorable A, B, and L regions displayed with red colors. (**D**) Histogram plot of Ramachandran, the distribution of residues in favored, allowed, and disallowed regions of PRRSV-V-1~PRRSV-V-3. (**E**–**G**) The Z-score plot of PRRSV-V-1 was scored as −5.98, PRRSV-V-1 score −4.8, and PRRSV-V-3 score as −1.24. (**H**) Histogram plot of Z-score of PRRSV-V-1~3, according to their Z-score plots.

**Figure 6 vetsci-12-00577-f006:**
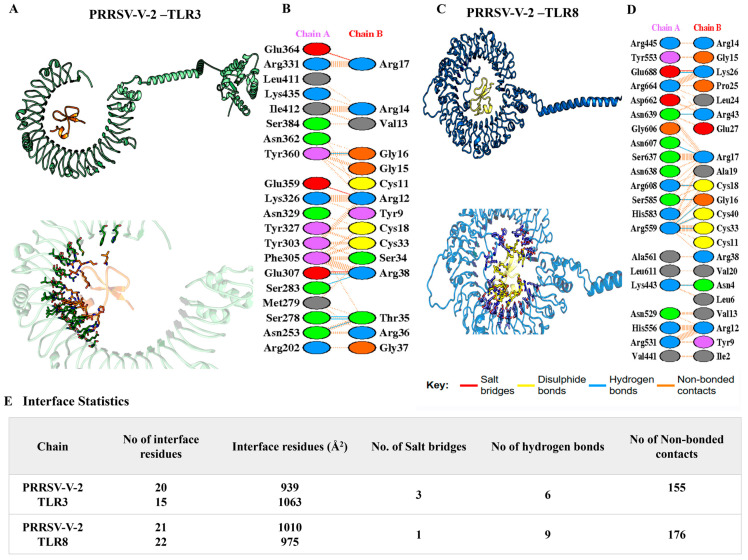
A molecular docking analysis of PRRSV-V-2-TLR3/8 complexes. PRRSV-V-2 docking with (**A**) TLR3 and (**C**) TLR8. The salt bridges (red), hydrogen bonds (blue), disulfide bonds (yellow), and non-bonded contacts (brown dots) formed in the interface between PRRSV-V-2 with (**B**) LR3 and (**D**) TLR8. (**E**) An analysis of the interface statistics properties of PRRSV-V-2 with TLR3 and TLR8.

**Figure 7 vetsci-12-00577-f007:**
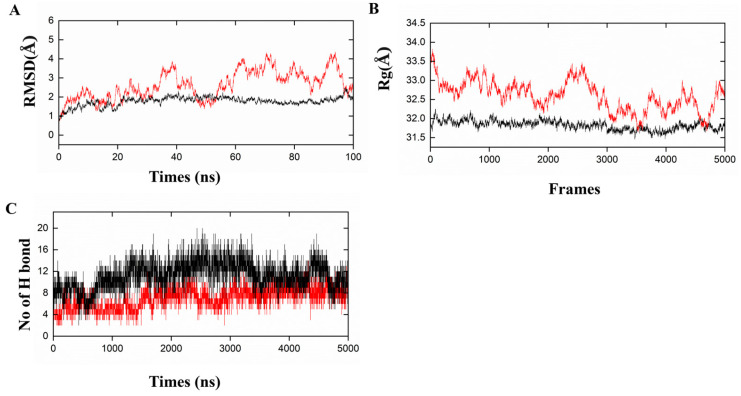
Molecular dynamic simulations of PRRSV-V-2 with TLR3/TLR8. (**A**) Root mean square deviation (RMSD): shows the RMSD for the complexes of PRRSV-V-2 with TLR3 and TLR8. (**B**) Radius of Gyration (Rg): Rg is the compactness of PRRSV-V-2 with TLR3 and TLR8. (**C**) Hydrogen Bonds: the count of intra-molecular hydrogen bonds formed by PRRSV-V-2 with TLR3 and TLR8. Graphical representations of the dynamic simulation of PRRSV-V-2 with TLR3 are indicated in red, whereas the dynamics simulation of PRRSV-V-2 with TLR8 is shown in black.

**Figure 8 vetsci-12-00577-f008:**
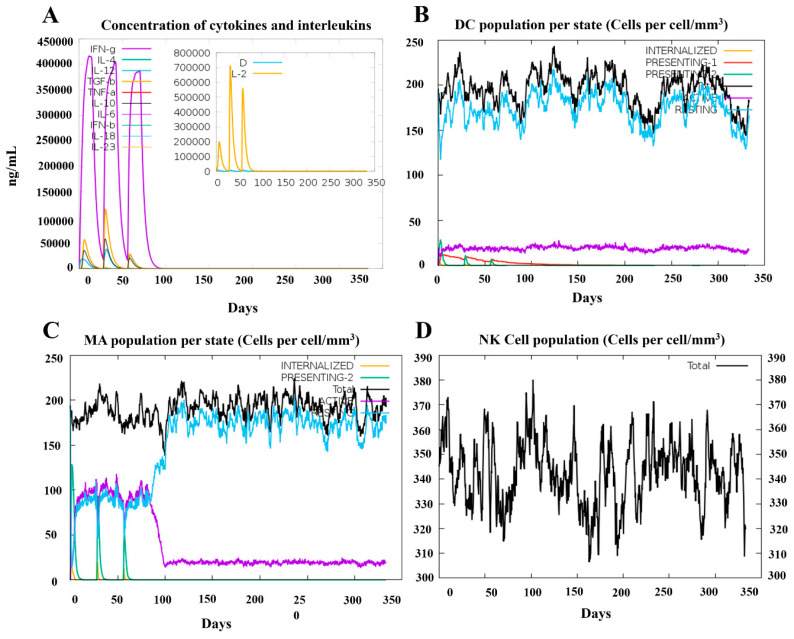
Simulation of innate immune response spots PRRSV-V-2 vaccination. (**A**) Kinetics of important cytokine expression levels after three PRRSV-V-2 dosages. (**B**) Dendritic cell (DC) responses following three PRRSV-V-2 doses. (**C**) Macrophages (MA) reactions to three doses of PRRSV-V-2 vaccination. (**D**) Natural killer (NK) cell dynamics after vaccination.

**Figure 9 vetsci-12-00577-f009:**
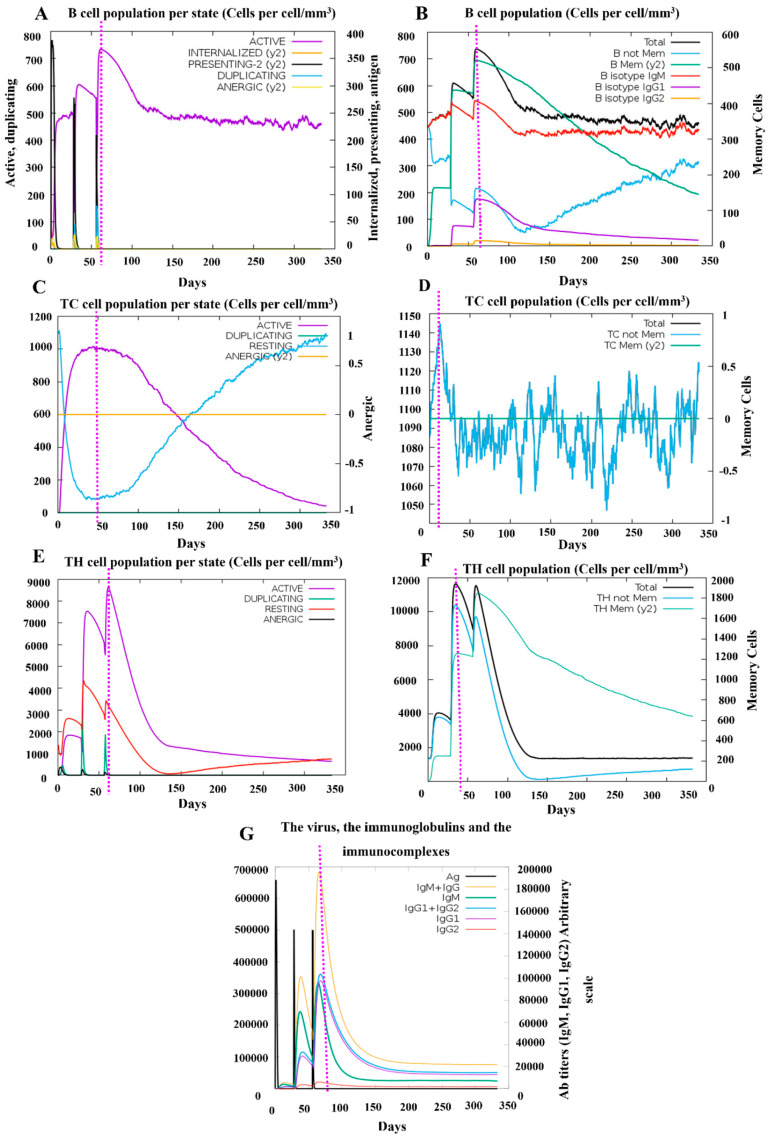
Simulation of the adaptive immune response following PRRSV-V-2 vaccination. (**A**) The status of the B-cell population post-vaccination is represented. (**B**) Distribution of B-cell subpopulations based on the antibody production status, (**C**,**D**) Dynamics of CD8+ cytotoxic T cells post-vaccination is shown, with distinct activation statuses (**C**), Such as active, duplicating, resting, (**D**) anergic (y2), and memory conditions. (**E**,**F**) Post-immunization trend CD4+ helper T cells are displayed, categorized by activation status (**E**), such as active, duplicating, resting, and (**F**) energic (y2), and memory conditions. (**G**) Immunoglobulin production following three PRRSV-V-2 doses shows antigen (Ag) levels.

**Table 1 vetsci-12-00577-t001:** Prediction of B-cell epitopes that are antigenic, non-allergic, and non-toxic.

Genotype	Protein	Start –End	Epitopes	Length	Antigenicity Score	Allergenicity /Toxicity	Intra/Inter Conservancy
PRRSV1	ORF2A	57–66	APRYSVRALP	10	0.8784	Non/Non	75.00% /20.00%
PRRSV2	GP3	38–45	LVRGNFSF	8	1.7037	Non/Non	100.00% /33.33%
PRRSV2	GP4	44–57	QHTQQHHLVIDHIR	14	0.5457	Non/Non	100.00% /33.33%
PRRSV1	GP5	49–61	LNGTDWLDKRFDW	13	1.7333	Non/Non	75.00% /20.00%
PRRSV2	M	131–144	RKPGLTSVNGTLV	13	0.8659	Non/Non	100.00% /33.33%
PRRSV1	N	6–13	HFPLAAED	8	0.9417	Non/Non	66.00% /22.22%

**Table 2 vetsci-12-00577-t002:** Prediction of HTL epitopes: Predicted antigenicity, allergenicity, interferon induction, toxicity, and conservancy analysis of shortlisted HTL (MHC-II) epitopes.

Genotype	Protein	HLA- Allele	Epitope	Antigenicity	Allergenicity/Toxicity	IL-4/IFN- γ Induction	Intra/Inter Conservancy
PRRSV1	ORF1a	DPA1*02:01 DPB1*05:01	GVAPAVRIAERYRGR	0.9596	Non/Non	Inducer /Inducer	100.00% /13.33%
PRRSV1	ORF1a	DRB4*01:04	KPIAYANLDEKKISA	0.9123	Non/Non	Inducer /Inducer	100.00% /13.33%
PRRSV2	ORF1b	DRB1*12:04	CLGDFKQLHPVGFDS	1.2326	Non/Non	Inducer /Inducer	95.24% /20.00%
PRRSV2	GP2a	DPA1*02:01 DPB1*04:01	LSFASDWFAPRYSVR	0.5461	Non/Non	Inducer /Inducer	75.00% /20.00%
PRRSV2	GP2b	DRB1*04:04	VFCIRLVCSAIHRSR	1.3347	Non/Non	Inducer /Inducer	100.00% /20.00%
PRRSV2	GP3	DPA1*02:01 DPB1*04:01	NWFHLEWLRPFFSSW	0.4583	Non/Non	Inducer /Inducer	75.00% /20.00%
PRRSV1	GP4	DRB5*01:03	ACVNFTDYVAHVTQH	1.0257	Non/Non	Inducer /Inducer	75.00% /13.33%
PRRSV2	GP5	DPA1*01:03 DPB1*01:01	WRYSCTRYTNFLLDT	0.5114	Non/Non	Inducer /Inducer	100.00% /13.33%
PRRSV1	M	DPA1*02:01 DPB1*02:01	LAFSITYTPIIYALK	1.4017	Non/Non	Inducer /Inducer	100.00% /13.33%
PRRSV2	N	DQA1*01:01 DQB1*02:01	PHFPLATEDDVRHHF	0.4807	Non/Non	Inducer /Inducer	66.67% /26.67%

**Table 3 vetsci-12-00577-t003:** Prediction of CTL epitopes. Shortlisted CTL (MHC-I) epitopes along with their strain, predicted immunogenicity, antigenicity, toxicity, and conservancy analysis.

Genotype	Protein	SLA-Allele	Epitope	Antigenicity /Immunogenicity	Allergenicity /Toxicity	Intra/Inter Conservancy
PRRSV1	ORF1a	1*1101	AALTGRTL	1.2071 /0.14994	Non/Non	100.00% /25.00%
PRRSV1	ORF1a	3*0501	HQKPIAYANL	1.1782 /0.1123	Non/Non	100.00% /20.00%
PRRSV2	ORF2b	3*0701	TRARHAIF	1.0977 /0.31383	Non/Non	95.24% /25.00%
PRRSV2	GP2a	3*0401	SVRALPFTL	1.5046 /0.15641	Non/Non	75.00% /22.22%
PRRSV1	GP2b	2*0101	IFLAILFGF	0.7123 /0.27353	Non/Non	75.00% /22.22%
PRRSV2	GP3	2*0101	YAWLAFLSF	1.1215 /0.10302	Non/Non	100.00% /22.22%
PRRSV2	GP4	1*0702	SACVNFTDY	1.6035 /0.17366	Non/Non	66.67% /22.22%
PRRSV2	GP5	3*0101	TRYTNFLL	1.2753 /0.1324	Non/Non	66.67% /25.00%
PRRSV1	M	2*1201	FSITYTPII	1.4228 /0.1836	Non/Non	66.67% /22.22%
PRRSV1	N	2*1101	FPLATEDDVRHHF	0.4645 /0.34939	Non/Non	50.00% /38.46%

**Table 4 vetsci-12-00577-t004:** Properties of PRRSV-V-1~3 vaccines in physiochemistry and immunology.

Parameters	PRRSV-V-1 Vaccine	PRRSV-V-2 Vaccine	PRRSV-V-3 Vaccine
**No. of amino acids**	531	448	551
**Molecular weight**	57,526.93 kDa	49,446.89 kDa	60,733.93 kDa
**Instability index**	22.04	25.72	28.66
**Aliphatic index**	81.02	80.91	77.04
**Half-life**	30 h (mammalian reticulocytes, in vitro). >20 h (yeast, in vivo). >10 h (Escherichia coli, in vivo)	30 h (mammalian reticulocytes, in vitro). >20 h (yeast, in vivo). >10 h (Escherichia coli, in vivo)	30 h (mammalian reticulocytes, in vitro). >20 h (yeast, in vivo). >10 h (Escherichia coli, in vivo)
**Solubility**	0.979905	0.975094	0.986037
**GRAVY**	−0.054	−0.191	−0.249
**Theoretical pI**	9.12	9.82	9.17
**Antigenicity**	0.61	0.68	0.63
**Allergenicity**	Non-allergenic	Non-allergenic	Non-allergenic
**Toxicity**	Non-toxic	Non-toxic	Non-toxic

**Table 5 vetsci-12-00577-t005:** Molecular docking and interaction analysis of PRRSV-V-1~PRRSV-V-3 with swine TLR2, TLR3, TLR4, and TLR8 complexes. Each vaccine–receptor complex is summarized by docking scores, the number of interface residues, interface area, salt bridges, hydrogen bonds, and non-covalent interactions.

Vaccine–Receptor Complex	Docking Score	Number of Residues in Interface	Area of Interface (Å2)	Number of Salt Bridges	Number of Hydrogen Bonds	Number of Non-Covalent Interactions
PRRSV-V-1 TLR-3	−196.74	15 20	1104 1114	2	3	121
PRRSV-V-1 TLR-8	−190.43	10 14	791 870	0	2	87
PRRSV-V-2 TLR-3	−308.16	20 15	939 1063	3	6	155
PRRSV-V-2 TLR-8	−263.17	21 22	1010 975	1	9	176
PRRSV-V-3 TLR-3	−260.99	30 37	2129 1943	4	3	316
PRRSV-V-3 TLR-8	−335.61	34 37	2340 2211	1	3	282

## Data Availability

All data are available in the manuscript and Appendix A.

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
