# Peer review of "In Silico Designed Multi-Epitope Vaccine Based on the Conserved Fragments in Viral Proteins for Broad-Spectrum Protection Against Porcine Reproductive and Respiratory Syndrome Virus"

_vetsci, 2025, doi:10.3390/vetsci12060577_

Round 1
Reviewer 1 Report
Comments and Suggestions for Authors
While the manuscript presents a good application of reverse vaccinology and immunoinformatics, it may require improvement in several areas before it can be considered for publication. Specifically:
- The manuscript lacks experimental validation or even an in vitro assessment of the predicted vaccine constructs. Although in silico approaches provide valuable insights into vaccine design, the claims regarding immunogenicity, safety, and efficacy remain speculative in the absence of empirical evidence. The authors should therefore revise their conclusions and explicitly acknowledge this limitation in both the abstract and discussion.
- Several of the bioinformatic tools and thresholds applied in the epitope selection and vaccine evaluation methods lack appropriate citations or justification. The authors should either provide relevant references or offer a rationale for their chosen parameters.
- The docking results are presented predominantly through binding energy values, without benchmarking against known ligand–receptor interactions or including appropriate controls. Likewise, the molecular dynamics simulation analysis reports only root mean square deviation (RMSD), root mean square fluctuation (RMSF), and radius of gyration values without including their biological relevance or linking these metrics to vaccine stability or receptor-binding potential in vivo.
Author Response
Reviewer Comment 1:
Comments and Suggestions for Authors
While the manuscript presents a good application of reverse vaccinology and immunoinformatics, it may require improvement in several areas before it can be considered for publication. Specifically:
- The manuscript lacks experimental validation or even an in vitro assessment of the predicted vaccine constructs. Although in silico approaches provide valuable insights into vaccine design, the claims regarding immunogenicity, safety, and efficacy remain speculative in the absence of empirical evidence. The authors should therefore revise their conclusions and explicitly acknowledge this limitation in both the abstract and discussion.
Response:
We appreciate the reviewer’s valuable observation. We fully acknowledge that the present study is based solely on in silico analyses and lacks in vitro or in vivo experimental validation. While computational tools offer powerful and efficient ways to predict vaccine potential, we recognize that claims regarding immunogenicity, safety, and efficacy remain predictive and must be interpreted cautiously.
Accordingly, we have revised both the Abstract and Discussion sections to state this limitation clearly. In the Abstract, we have mentioned that the findings are based on computational predictions and require experimental validation (see lines 49-52). In the Discussion, we explicitly acknowledge that further in vitro and in vivo studies are necessary to confirm the vaccine construct’s biological performance before considering any clinical application (see lines 636-648, page 21). This clarification helps ensure that our conclusions remain balanced and grounded in the scope of our methodology.
- Several bioinformatic tools and thresholds applied in the epitope selection and vaccine evaluation methods lack appropriate citations or justification. The authors should either provide relevant references or offer a rationale for their chosen parameters.
Response:
Thank you for your valuable feedback. We have thoroughly revised the Methods section—particularly Sections 2.2 and 2.4—to incorporate detailed citations, clearly explain threshold selections, and reference validation sources for all immunoinformatics tools employed in epitope prediction and vaccine candidate evaluation. Below is a breakdown of the updated methodological approach and justifications:
- Linear B-cell epitopes were predicted using the BepiPred-2.0 tool available on the IEDB platform (http://tools.iedb.org/bepipred/). A threshold score of >0.5 was applied, as recommended in the original publication by Jespersen et al. (2017) (Reference 1; Section 2.2, Line 160).
- HTL (MHC class II) epitopes were predicted using NetMHCIIpan, applying a top 10 % percentile rank (%Rank ≤10% or IC50 <1000 nM) threshold (Reference 2). To ensure relevance to immune modulation, we retained only those epitopes predicted to induce both IFN-γ (via IFNepitope) (Reference 8) and IL-4 (via IL4pred) (Reference 9).
- CTL (MHC class I) epitopes were predicted using the NetMHCpan EL method (latest version, as of September 2023), also available via IEDB. We applied a standard binding affinity cutoff of the top 1% of percentile rank (%Rank ≤1% or IC50 <500 nM) to identify strong binders, following IEDB guidelines and validated literature (Reference 2). Further filtering was conducted using the IEDB Class I Immunogenicity tool, and only epitopes with immunogenicity scores >0.2 were retained, excluding peptides predicted as non-immunogenic (Reference 4).
- Additionally, our selection criteria align with established interpretations of binding strength:
- HTL/MHC II epitopes: Strong Binders (SB) when %Rank <2%; Weak Binders (WB) when %Rank <10% (Section 2.2, Line 166; References 2, 3).
- CTL/MHC I epitopes: SB when %Rank <0.5%; WB when %Rank <2% (Section 2.2, Line 173; References 2, 3).
- All predicted epitopes (B-cell, HTL, CTL) underwent further screening for:
- Antigenicity via VaxiJen v2.0 (threshold: >0.4 for viral antigens) (Reference 5),
- Allergenicity via AllerTOP v2.0 (Reference 6), and
- Toxicity via ToxinPred (Reference 7).
Only epitopes meeting all the following criteria were retained: ≥50% sequence identity, antigenicity score >0.5, non-allergenic, and non-toxic. Epitopes failing to meet these criteria—i.e., those identified as non-antigenic, allergenic, or toxic—were excluded from the final vaccine construct. Finally, our strategy was based comprehensive ranking strategy was applied to prioritized epitopes with the highest immunogenic potential for the vaccine construct:
- For CTL epitopes: ranking incorporated %Rank, antigenicity, and immunogenicity scores.
- For HTL epitopes, ranking was based on %Rank and antigenicity scores.
This weighted scoring framework ensured the selection of epitopes with strong binding affinity, high antigenicity, and favorable immunogenic profiles.
Note: All supporting studies referenced in this section are also cited in the manuscript as Reference Nos. 47, 49, 51, 52, 53, 54, 55, 56, and 82.
References
- Jespersen MC, Peters B, Nielsen M, Marcatili P. 2017. BepiPred-2.0: improving sequence-based B-cell epitope prediction using conformational epitopes. Nucleic Acids Res (Web Server issue).2:2.PMID: 28472356
- Reynisson B, Alvarez B, Paul S, Peters B, Nielsen M. NetMHCpan-4.1 and NetMHCIIpan-4.0: improved predictions of MHC antigen presentation by concurrent motif deconvolution and integration of MS MHC eluted ligand data. Nucleic Acids Res. 2020 Jul 2;48(W1):W449-W454. doi: 10.1093/nar/gkaa379. PMID: 32406916; PMCID: PMC7319546.
- Liu D, Chen Y. Epitope screening and vaccine molecule design of PRRSV GP3 and GP5 protein based on immunoinformatics. J Cell Mol Med. 2024 Feb;28(3):e18103. doi: 10.1111/jcmm.18103. Epub 2024 Jan 12. PMID: 38217314; PMCID: PMC10844699.
- Calis, J.J., et al., Properties of MHC class I presented peptides that enhance immunogenicity. PLoS computational biology, 2013. 9(10): p. e1003266.
- Doytchinova, I.A. and D.R. Flower, VaxiJen: a server for prediction of protective antigens, tumour antigens and subunit vaccines. BMC bioinformatics, 2007. 8: p. 1-7.
- Dimitrov, I., et al., AllerTOP v. 2—a server for in silico prediction of allergens. Journal of molecular modeling, 2014. 20: p. 1-6.
- Gupta, S., et al., In silico approach for predicting toxicity of peptides and proteins. PloS one, 2013. 8(9): p. e73957.
- Dhanda, S.K., P. Vir, and G.P. Raghava, Designing of interferon-gamma inducing MHC class-II binders. Biology direct, 2013. 8: p. 1-15.
- Dhanda, S.K., et al., Prediction of IL4 Inducing Peptides. Journal of Immunology Research, 2013. 2013(1): p. 263952.
Further details are provided in the Methods section (Section 2.4, Line 205, Page 5), where we describe the evaluation of physicochemical properties using the ExPASy ProtParam tool. According to established thresholds, an instability index below 40 indicates a stable protein in line 207(Reference 1). All vaccine constructs—PRRSV-V-1, PRRSV-V-2, and PRRSV-V-3—exhibited instability index values of 22.04, 25.72, and 28.66, respectively, confirming their structural stability (Section 3.4, Line 3; Table 4, Page 10).
Solubility analysis was conducted using SOLpro from the SCRATCH server, where a solubility probability greater than 0.5 indicates a soluble protein (Section 2.4, Line 209, Page 5; Reference 2). The three vaccine constructs—PRRSV-V-1, PRRSV-V-2, and PRRSV-V-3—achieved solubility scores of 0.979905, 0.975094, and 0.986037, respectively. These values, all well above 0.9, suggest excellent solubility and strong potential for expression efficiency (Section 3.4, Line 6; Table 4, Page 11).
Note: The references supporting these analyses are cited in the manuscript as Reference Nos. 62 and 63.
Reference
- Gasteiger, E., et al., Protein Analysis Tools on the ExPASy Server 571 571 From: The Proteomics Protocols Handbook Protein Identification and Analysis Tools on the ExPASy Server. The Proteomics Protocols Handbook, 2019: p. 571-607.
- Magnan, C.N., A. Randall, and P. Baldi, SOLpro: accurate sequence-based prediction of protein solubility. Bioinformatics, 2009. 25(17): p. 2200-2207.
- The docking results are presented predominantly through binding energy values, without benchmarking against known ligand–receptor interactions or including appropriate controls. Likewise, the molecular dynamics simulation analysis reports only root mean square deviation (RMSD), radius of gyration values (Rg), and number of hydrogen bonds without including their biological relevance or linking these metrics to vaccine stability or receptor-binding potential in vivo.
Response:
Thank you for this valuable observation. We acknowledge the importance of benchmarking in validating docking results. In our study, docking analyses were primarily assessed based on binding energy values, following methodologies reported in peer-reviewed vaccine design literature, where the lowest-energy (most negative) conformations are commonly interpreted as indicative of favorable receptor–ligand interactions. Specifically, we employed the HDock server and selected top-ranked models based on minimum binding energy, consistent with established computational vaccinology practices.
While benchmarking against known experimental ligand–receptor complexes was not explicitly included due to the unavailability of such reference complexes for PRRSV vaccine constructs, our docking pipeline closely mirrors widely accepted protocols in the field. In addition to docking scores, we performed detailed structural analyses, including interface residue mapping, hydrogen bonding patterns, and surface complementarity assessments, to reinforce the biological plausibility of the predicted complexes. A similar approach using the HDOCK server for docking and selection of top-ranked models based on minimum binding energy has been adopted in previous studies, such as Reference 1 and Reference 2 in their multi-epitope vaccine design, thereby reinforcing the validity of our methodology
References
1 Beig M, Sholeh M, Moradkasani S, Shahbazi B, Badmasti F. Development of a multi-epitope vaccine against Acinetobacter baumannii: A comprehensive approach to combating antimicrobial resistance. PLoS One. 2025 Mar 10;20(3):e0319191. doi: 10.1371/journal.pone.0319191. PMID: 40063635; PMCID: PMC11892874.
2 Asad M, Hassan A, Wang W, Alonazi WB, Khan MS, Ogunyemi SO, Ibrahim M, Bin L. An integrated in silico approach for the identification of novel potential drug target and chimeric vaccine against Neisseria meningitides strain 331401 serogroup X by subtractive genomics and reverse vaccinology. Comput Biol Med. 2024 Aug;178:108738. doi: 10.1016/j.compbiomed.2024.108738. Epub 2024 Jun 10. PMID: 38870724.
We thank the reviewer for the insightful suggestion. In the revised Results section 3.6 431-452, page 15, we have clarified the reporting of MD simulation metrics and provided their biological relevance in a separate paragraph.
- RMSD: The RMSD values for PRRSV-V-2–TLR3 and TLR8 complexes remained consistently about 5 Å, indicating structural stability and convergence (Figure 6A).
- Radius of Gyration (Rg): Rg values remained stable between 32.5–33.5 Å throughout the simulation, suggesting no major conformational fluctuations (Figure 6B).
- Hydrogen Bonds: A higher and consistent number of hydrogen bonds were observed in PRRSV-V-2 complexes, indicating persistent intermolecular interactions (Figure 6C).
To address the reviewer’s concern, we have also added a dedicated paragraph discussing the biological implications of these findings, highlighting how these parameters relate to complex stability, antigen presentation, and potential immunogenicity in vivo in the last paragraph of the Discussion part, line579-587
Reference:
Ullah H, Ullah S, Li J, Yang F, Tan L. An In Silico Design of a Vaccine against All Serotypes of the Dengue Virus Based on Virtual Screening of B-Cell and T-Cell Epitopes. Biology (Basel). 2024 Aug 30;13(9):681. doi: 10.3390/biology13090681. PMID: 39336108; PMCID: PMC11428656

Reviewer 2 Report
Comments and Suggestions for Authors
The work presents an interesting developed a novel multi-epitope vaccine to address PRRSVʹs high mutation rate and genetic variability. Also, the authors present a promising strategy for developing a broadly protective PRRSV vaccine but also offers a new approach to improving disease control in the swine industry, mitigating the impact of diverse viral strains.
I have the following observations and/or suggestions:
In Figure 1, the order of the information in columns 1 and 3 is correct. It is not clear how the phylogenetic tree results allow for the generation of conserved fragments. In column 3, it is not clear how the molecular dynamics results allow for the simulation of immunoreactions.
On page 5, please indicate the criteria for using the PHYRE2 server and what advantages does it offer over other strategies or programs?
Page 5. In the molecular dynamics section, why use the FF14SB force field instead of CHARMM or Amber?
Page 6. Why run molecular dynamics at 300 K (26.87 °C)? This is the temperature usually defined for the analysis, but it is far from cellular conditions.
Page 6. 3. Results. What database in the NCBI domain were the 7315 sequences extracted from?
On page 8, there is a Table 3 label, before the Table 2 label, which should be relocated to its respective Table 3.
In Table 5, I suggest considering confirming the theoretical half-lives of the analyzed proteins with another program. How can we explain that they have the same half-life in the yeast, E. coli, and reticulocyte models?
I suggest moving Figure 9 to the supplementary material.
Author Response
Reviewer Comment 2:
Comments and Suggestions for Authors
The work presents an interesting development of a novel multi-epitope vaccine to address PRRSVʹs high mutation rate and genetic variability. Also, the authors present a promising strategy for developing a broadly protective PRRSV vaccine but also offer a new approach to improving disease control in the swine industry, mitigating the impact of diverse viral strains.
I have the following observations and/or suggestions:
In Figure 1, the order of the information in columns 1 and 3 is correct. It is not clear how the phylogenetic tree results allow for the generation of conserved fragments. In column 3, it is not clear how the molecular dynamics results allow for the simulation of immunoreactions.
Response:
While the sequential organization of information in columns 1 and 3 is appropriate, a correction is warranted regarding column 1. The depiction of a phylogenetic tree was mistakenly included; in actuality, conserved fragment identification was achieved through multiple sequence alignment (MSA) of target protein sequences derived from diverse viral strains. The MSA enabled the detection of evolutionarily conserved regions, which were subsequently extracted as candidate antigenic fragments for epitope mapping. This approach ensures that the vaccine design captures broadly conserved elements, potentially conferring cross-protection against multiple viral variants.
In column 3, the role of molecular dynamics (MD) simulations should be contextualized more precisely regarding immunogenicity. MD simulations do not directly replicate immune responses but provide critical insights into the biophysical behavior of vaccine–immune receptor complexes, specifically their structural stability, conformational adaptability, and intermolecular interactions over time. These parameters serve as indirect indicators of immunogenic potential, particularly when evaluating the capacity of vaccine constructs to form stable and persistent interactions with pattern recognition receptors such as Toll-like receptors (TLRs). Following MD simulations, in silico immune profiling is conducted using computational platforms such as C-ImmSim. These simulations predict host immune responses by modeling the activation kinetics of B-cells, T-cells, cytokine secretion patterns, and immunological memory development, offering a comprehensive evaluation of the vaccine’s prospective immunoprotective efficacy. The revised Figure 1 is shown below and can also be found in the Methods section, Figure 1, page 4 of the manuscript
On page 5, please indicate the criteria for using the PHYRE2 server and what advantages it offers over other strategies or programs.
Response: Thank you for your valuable feedback. In response to your inquiry regarding the criteria for using the PHYRE2 server and its advantages over other strategies or programs, we would like to provide the following clarification:
Criteria for Using PHYRE2:
To initiate the structural modeling process in the PHYRE2 server, a protein sequence in FASTA format is required as input. The quality and accuracy of modeling largely depend on the quality of the input sequence and its similarity to known structures deposited in the Protein Data Bank (PDB). Therefore, we recommend ensuring the sequence is well-formatted, free from errors, and preferably homologous to known structures to improve model prediction reliability.
Advantages of PHYRE2 Over Other Methods:
- Advanced Threading-Based Approach: Unlike conventional homology-based methods (RaptorX, TrRosseta), PHYRE2 employs threading algorithms, also known as fold recognition, to map the query sequence onto templates similar in structure. This enables accurate structural predictions, particularly for proteins with low sequence identity (below 40%) to known structures, where traditional alignment methods may fail.
- Comprehensive Template Database: PHYRE2 utilizes an extensive and up-to-date template database, allowing it to find the most relevant and similar protein structures for homology modeling. This large repository of templates significantly enhances its ability to generate reliable models, making it an advantage over other tools like Swiss-Model and I-TASSER, which rely on slightly smaller databases.
- High Model Accuracy: The hybrid nature of PHYRE2, which integrates multiple modeling techniques (including homology modeling, ab initio methods, and threading), ensures more accurate and robust structural predictions compared to single-method servers. This hybridization is particularly beneficial for proteins that do not have high sequence homology with available templates.
- Integrated Structural Refinement and Validation: In addition to generating initial models, PHYRE2 provides tools for model refinement and structural validation, making it possible to improve and assess the quality of the predicted structures. Furthermore, it integrates with visualization tools like PyMOL to help interpret the structural data.
- Confidence Scores: One of the standout features of PHYRE2 is its ability to generate confidence scores for each predicted model, providing users with insight into the reliability of the structure. These confidence metrics help prioritize which models are most likely to be accurate, which is crucial for downstream analyses.
In summary, PHYRE2 offers a powerful and flexible platform for protein structure prediction, particularly in cases where sequence similarity to known structures is low. Its threading algorithms, extensive template database, model refinement capabilities, and confidence scoring system set it apart from other programs and make it a preferred choice for many researchers in structural bioinformatics. Especially, since our current work is focusing on the in-silico designing of a multi-epitope vaccine, the virtual molecule generated was assembled with a panel of fragments extracted from natural protein, which means a low similarity to a certain known structure. Fortunately, this property just matches the advantage of PHYRE2. For this reason, we chose PHYRE2 as the structure modeling tool in our study.
Correspondingly, some statements were added in “material and method” as a brief introduction of PHYRE2 and to show the reason we adopted this server. Please see Line 217 to Line 222, Page 05.
Page 5. In the molecular dynamics section, why use the FF14SB force field instead of CHARMM or Amber?
Response:
Thank you for the question. FF14SB is the force field of AMBER. We forgot to mention it in the manuscript, but now it has been corrected. We employed the Amber FF14SB force field because it is specifically optimized for protein and peptide systems, providing improved backbone and side-chain torsion angle parameters over earlier Amber force fields. FF14SB has been extensively validated for protein folding, peptide dynamics, and protein–protein interactions, making it a reliable choice for vaccine–receptor complex simulations. The correct statement can be seen line “material and method”, please see Line 244, Page 06.
Page 6. Why run molecular dynamics at 300 K (26.87 °C)? This is the temperature usually defined for the analysis, but it is far from cellular conditions.
Thank you for the question. We conducted molecular dynamics simulations at 300 K because this temperature is a standard reference point at which the AMBER FF14SB force field and TIP3P water model have been extensively validated. Increasing the temperature above 300 K would elevate the system’s kinetic energy, potentially leading to excessive molecular motion and delayed equilibration, thereby reducing simulation stability. Moreover, many experimental studies performed in vitro operate at or near room temperature, which supports the relevance of 300 K for simulating stable structural behavior. Running simulations at 300 K ensures reliable thermodynamic properties, reproducible dynamics, and comparability with numerous published benchmark studies that also use this temperature (REFERENCE 1 and 2).
If we were to increase the temperature to 310 K (approximately 37°C, close to physiological conditions), the system’s molecular motion would become more pronounced, and equilibration may take longer due to the higher energy state. While this temperature could better represent physiological conditions, it could also lead to an overestimation of flexibility or induce non-physiological conformations due to increased thermal motion. This would require additional computational time for the system to stabilize, and the potential for higher variability in results could compromise the accuracy of our predictions. Given these factors, we opted for 300 K as a more balanced and widely accepted temperature for simulations of this nature, which allows for stable results while minimizing unnecessary complexity in the simulation setup.
References
1 Allemailem, K. S. (2021). A comprehensive computer-aided vaccine design approach to propose a multi-epitope subunit vaccine against the genus Klebsiella using pan-genomics, reverse vaccinology, and biophysical techniques. Vaccines, 9(10), 1087.
2 Hazra D, Rahman S, Ganguly M, Das AK, Roychowdhury A. Molecular dynamics simulation shows enhanced stability in scaffold-based macromolecule, designed by protein engineering: a novel methodology adapted for converting Mtb Ag85A to a multi-epitope vaccine. J Mol Model. 2025 Feb 15;31(3):84. doi: 10.1007/s00894-025-06301-2. PMID: 39954152.
Page 6. 3. Results. What database in the NCBI domain were the 7315 sequences extracted from?
Response: Thank you for your question regarding the source of the 7315 sequences used in our study. The sequences were extracted from the NCBI domain, specifically from the NCBI protein databases (https://www.ncbi.nlm.nih.gov/protein) with keywords protein names of PRRSV, like GP2, GP3, and so on. These databases provide a comprehensive collection of protein sequences from a wide variety of organisms and are routinely used in genomic research to ensure the accuracy and relevance of the data.
On page 8, there is a Table 3 label, before the Table 2 label, which should be relocated to its respective Table 3.
Response: Thank you for pointing out the labeling issue with the tables in our manuscript. We have corrected the error where Table 3 was incorrectly labeled before Table 2. The label has now been relocated to its respective Table 3 on page 08. Furthermore, we have inspected the labels of Tables and Figures in our manuscript, and all of them have been ensured in the correct location. We appreciate your attention to detail and apologize for any confusion this may have caused.
In Table 5, I suggest confirming the theoretical half-lives of the analyzed proteins with another program. How can we explain that they have the same half-life in the yeast, E. coli, and reticulocyte models?
Response: Thank you for your question regarding the tool used to predict the half-lives of the analyzed proteins. In our study, we employed the ProtParam tool to estimate the theoretical half-lives of the vaccine candidate proteins. ProtParam bases its predictions on the N-end rule, a well-established model that correlates protein half-life with the identity of the N-terminal amino acid residue. This model is supported by experimental observations indicating that specific N-terminal residues are preferentially recognized by host degradation pathways, thereby influencing protein stability.
We fully recognize the value of validating computational predictions with alternative tools. However, to the best of our knowledge, there are currently no other publicly available tools that offer theoretical half-life predictions across multiple host systems (e.g., E. coli, yeast, reticulocytes) based solely on protein sequence. As such, ProtParam remains the widely accepted standard for theoretical half-life estimation in computational protein design and vaccinology.
It is important to note that ProtParam assumes conserved degradation mechanisms across model organisms. Therefore, when protein constructs share the same N-terminal residue, the tool will yield identical half-life predictions for all expression systems analyzed. This explains the uniformity observed in our results. Additionally, recent literature has employed similar approaches for theoretical half-life prediction using ProtParam, with reported results consistent with our methodology. Theoretical half-life estimates comparable to their Amino Acids can also be found in Table 4 of Reference 3, which further supports the reliability of our approach.
Table of the amino acids and the corresponding half-life (https://web.expasy.org/protparam/protparam-doc.html)
|
Amino acid |
Mammalian |
Yeast |
E. coli |
|
Ala |
4.4 hour |
>20 hour |
>10 hour |
|
Arg |
1 hour |
2 min |
2 min |
|
Asn |
1.4 hour |
3 min |
>10 hour |
|
Asp |
1.1 hour |
3 min |
>10 hour |
|
Cys |
1.2 hour |
>20 hour |
>10 hour |
|
Gln |
0.8 hour |
10 min |
>10 hour |
|
Glu |
1 hour |
30 min |
>10 hour |
|
Gly |
30 hour |
>20 hour |
>10 hour |
|
His |
3.5 hour |
10 min |
>10 hour |
|
Ile |
20 hour |
30 min |
>10 hour |
|
Leu |
5.5 hour |
3 min |
2 min |
|
Lys |
1.3 hour |
3 min |
2 min |
|
Met |
30 hour |
>20 hour |
>10 hour |
|
phe |
1.1 hour |
3 min |
2 min |
|
Pro |
>20 hour |
>20 hour |
? |
|
Ser |
1.9 hour |
>20 hour |
>10 hour |
|
Thr |
1.7 hour |
>20 hour |
>10 hour |
|
Trp |
2.8 hour |
3 min |
2 min |
|
Tyr |
2.8 hour |
10 min |
2 min |
|
Val |
100 hour |
>20 hour |
>10 hour |
References
1 Gonda, D.K., Bachmair, A., Wunning, I., Tobias, J.W., Lane, W.S. and Varshavsky, A. J. (1989) Universality and structure of the N-end rule. J. Biol. Chem. 264, 16700-16712.
2 Guruprasad, K., Reddy, B.V.B. and Pandit, M.W. (1990). Correlation between stability of a protein and its dipeptide composition: a novel approach for predicting in vivo stability of a protein from its primary sequence. Protein Eng. 4,155-161.
3 Ullah H, Ullah S, Li J, Yang F, Tan L. An In Silico Design of a Vaccine against All Serotypes of the Dengue Virus Based on Virtual Screening of B-Cell and T-Cell Epitopes. Biology (Basel). 2024 Aug 30;13(9):681. doi: 10.3390/biology13090681. PMID: 39336108; PMCID: PMC11428656.
I suggest moving Figure 9 to the supplementary material.
Response: Thank you for your suggestion regarding Figure 9. We appreciate your recommendation to improve the clarity and flow of the main text. We agree that Figure 9, while informative, may be more appropriately placed in the Supplementary Material to maintain the focus of the main results section. Accordingly, we have moved Figure 9 to the supplementary section and updated all in-text references to reflect this change, and shown in the supplementary section, Figure S8.

Reviewer 3 Report
Comments and Suggestions for Authors
In this paper, the authors present a promising strategy for developing a broadly protective PRRSV vaccine. They evaluated the vaccines in silico closely, such as physiochemistry properties, structural stability, binding with pattern recognition receptors, and induction of the host immune system. Although these properties are important, it seems that TLR3 or TLR8 is unsuitable receptor for this study. It is also clear that the authors are not native speakers of English. They should maintain consistency in terminology usage (ex. both aa and amino acids are used).
Specific comments:
- Lines 38 and 44. CTLs, HTLs, and TLR should be spelled out.
- Line 48. PRRSV not PRRSV virus.
- Line 50. Reverse Vaccinology should be included in the Abstract or removed from the Keywords.
- Line 183. Mycobacterium tuberculosis should start with a capital letter.
- Lines 260 and 263. PRRSV1 and PRRSV2 are type 1 and type2, respectively?
- Table 1 and others. The title should start with a capital letter.
- Tables 2 and 3. Duplicate?
- Table 4 should be improved.
- Fig. 9. pPRRSV-V-2 should be used.
- Lines 570-571. This sentence is an overstatement.
The authors should maintain consistency in terminology usage.
Author Response
Reviewer Comment 3:
Comments and Suggestions for Authors
In this paper, the authors present a promising strategy for developing a broadly protective PRRSV vaccine. They evaluated the vaccines in silico closely, such as physicochemical properties, structural stability, binding with pattern recognition receptors, and induction of the host immune system. Although these properties are important, it seems that TLR3 or TLR8 is an unsuitable receptor for this study. It is also clear that the authors are not native speakers of English. They should maintain consistency in terminology usage (ex., both aa and amino acids are used).
Response:
Thank you for your thoughtful and constructive feedback. We appreciate your recognition of the in-silico evaluation strategy employed in our study.
Toll-like receptors (TLRs) are pivotal in the innate immune system's ability to detect pathogen-associated molecular patterns (PAMPs), and their engagement is critical for initiating robust adaptive immune responses. The inclusion of TLR-targeted docking in in-silico vaccine design has become a widely adopted strategy to enhance the predicted immunogenicity and functional efficacy of multiepitope constructs.
In particular, TLR2, TLR3, TLR4, TLR5, TLR7, TLR8, and TLR9 are commonly targeted in computational vaccinology, either through adjuvant design or direct docking simulations, due to their ability to recognize viral, bacterial, and other pathogenic components. Simulating interactions between designed vaccine constructs and these receptors allows for an in-depth evaluation of structural compatibility, binding stability, and immunostimulatory potential—key metrics in predicting the translational value of a vaccine candidate.
In our study, TLR3 and TLR8 were selected as representative innate immune receptors based on their well-documented roles in viral RNA recognition, interferon signaling, and pro-inflammatory cytokine production, especially in swine immunology. These receptors have been previously utilized in several swine-specific in silico vaccine design studies, which employed molecular docking and dynamic simulation to characterize and validate vaccine–host interactions.
To support the relevance of our approach, we have incorporated references to recent peer-reviewed literature where TLR3 and TLR8 were similarly used as receptor models for docking with viral vaccine constructs. These studies further validate our receptor selection and strengthen the rationale behind their inclusion in the docking simulations. The supporting references have now been cited appropriately in the revised manuscript as references 70 and 83.
Reference
1 Chen X, Li Y, Wang X. Multi-epitope vaccines: a promising strategy against viral diseases in swine. Front Cell Infect Microbiol. 2024 Dec 20;14:1497580. doi: 10.3389/fcimb.2024.1497580. PMID: 39760092; PMCID: PMC11695243.
2 Liu D, Chen Y. Epitope screening and vaccine molecule design of PRRSV GP3 and GP5 protein based on immunoinformatics. J Cell Mol Med. 2024 Feb;28(3):e18103. doi: 10.1111/jcmm.18103. Epub 2024 Jan 12. PMID: 38217314; PMCID: PMC10844699.
As for the language and terminology consistency, we have conducted a thorough revision of the manuscript for English clarity and scientific writing quality. All terminology has been standardized throughout the text—for example, replacing mixed usage of "aa" with "amino acids" consistently, to improve readability and professionalism.
Specific comments:
- Lines 38 and 44. CTLs, HTLs, and TLR should be spelled out.
Response:
Thank you for your attention to clarity and readability. In response, I have revised lines 38 and 44 to spell out the abbreviations upon first use: Helper T Lymphocytes (HTLs), Cytotoxic T Lymphocytes (CTLs), and Toll-like Receptors (TLRs). This ensures consistency with scientific writing conventions and improves accessibility for readers unfamiliar with the abbreviations. These changes have been incorporated in the updated version of the manuscript and can be found at Line 39 and Line 45.
- Line 48. PRRSV not PRRSV virus.
Response:
Thank you for pointing this out. The phrase at line 48 has been corrected to use "PRRSV" instead of the redundant term "PRRSV virus", under proper scientific nomenclature.
- Line 50. Reverse Vaccinology should be included in the Abstract or removed from the Keywords.
Response:
Thank you for your observation. In response, I have revised the manuscript to ensure consistency. The term "Reverse Vaccinology" has now been included in the Abstract in line 35, where the overall computational approach is summarized. This update aligns with its presence in the Keywords section and clarifies the methodological framework used in the study.
- Line 183. Mycobacterium tuberculosis should start with a capital letter.
Response:
Thank you for pointing this out. The species name at line 183 has been corrected to Mycobacterium tuberculosis, with proper capitalization and italicization according to scientific naming conventions. In the updated manuscript, Mycobacterium tuberculosis appears on line 193 of page 5.
- Lines 260 and 263. PRRSV1 and PRRSV2 are type1 and type2, respectively?
Response:
Thank you for your question. Yes, PRRSV1 and PRRSV2 refer to type 1 (European genotype) and type 2 (North American genotype), respectively. This distinction has now been clarified in the revised text of the updated manuscript at lines 280 and 283 to avoid ambiguity and improve the accuracy of the manuscript.
- Table 1 and others. The title should start with a capital letter.
Response:
Thank you for your observation. The titles of Table 1 and all other tables have been revised to begin with a capital letter, ensuring consistency with formatting standards throughout the manuscript.
Table 1, page 7
Table 2, page 8
Table 3, page 8
Table 4, page 10
Table 5, page 14
- Tables 2 and 3. Duplicate?
Response:
Thank you for your question. Table 2 and Table 3 are distinct from one another. Table 2 provides predictions for Helper T Lymphocyte (HTL) or MHC II, also called CD4+ T cell epitopes, along with their associated properties, such as antigenicity, allergenicity, toxicity, IL-4/IFN-γ induction, and conservancy. In contrast, Table 3 focuses on Cytotoxic T Lymphocyte (CTL) OR MHCI, also called CD8+ T cells, epitopes, and their properties, including antigenicity, allergenicity, toxicity, immunogenicity, and conservancy. These tables differ in the type of T-cell epitopes being predicted and the specific properties assessed for each.
- Table 4 should be improved.
Response:
Thank you for your feedback. Table 4 has been revised to enhance clarity and readability. The Table title has been modified, the unit of molecular weight (kDa) has been added to line 2, and revisions have been made for better accuracy. The updated version includes more precise column headers, consistent units and formatting, and, where applicable, brief annotations to aid interpretation. This enhancement ensures that the data is easier to follow and more effectively supports the results discussed in the manuscript.
- Fig. 9. pPRRSV-V-2 should be used.
Response:
Thank you for your valuable observation. Figure 9 has been revised to accurately label the construct as pPRRSV-V-2, ensuring consistency with the nomenclature used throughout the manuscript. Additionally, by the recommendation of another reviewer, the updated figure has been relocated to the supplementary section as Supplementary Figure S8.
- Lines 570-571. This sentence is an overstatement.
Response:
Thank you for pointing this out. The sentence at lines 570–571 has been revised to adopt a more measured tone, avoiding overstatement. The current sentence is “By using comprehensive in silico approaches, we designed a novel multi-epitope vaccine candidate to address PRRSV's high mutation rate and genetic variability”. The corrected statement can be found on Lines 620–621 of the updated manuscript. This revised sentence reflects that the work is computational and avoids implying experimental validation or confirmed efficacy.

Round 2
Reviewer 2 Report
Comments and Suggestions for Authors
I greatly appreciate the authors' kind response. They took the modification suggestions into account. I have no further comments.